# The Effects of Prenatal Supplementation with β-Hydroxy-β-Methylbutyrate and/or Alpha-Ketoglutaric Acid on the Development and Maturation of Mink Intestines Are Dependent on the Number of Pregnancies and the Sex of the Offspring

**DOI:** 10.3390/ani11051468

**Published:** 2021-05-20

**Authors:** Piotr Dobrowolski, Siemowit Muszyński, Janine Donaldson, Andrzej Jakubczak, Andrzej Żmuda, Iwona Taszkun, Karol Rycerz, Maria Mielnik-Błaszczak, Damian Kuc, Ewa Tomaszewska

**Affiliations:** 1Department of Functional Anatomy and Cytobiology, Institute of Biological Sciences, Faculty of Biology and Biotechnology, Maria Curie-Sklodowska University, Akademicka St. 19, 20-033 Lublin, Poland; 2Department of Biophysics, Faculty of Environmental Biology, University of Life Sciences in Lublin, Akademicka St. 13, 20-950 Lublin, Poland; siemowit.muszynski@up.lublin.pl; 3School of Physiology, Faculty of Health Sciences, University of the Witwatersrand, 7 York Road, Parktown, Johannesburg 2193, South Africa; Janine.Donaldson@wits.ac.za; 4Department of Biological Basis of Animal Production, Faculty of Biology and Animal Breeding, University of Life Sciences in Lublin, Akademicka St. 13, 20-950 Lublin, Poland; andrzej.jakubczak@up.lublin.pl; 5Department of Epizootiology and Clinic of Infectious Diseases, Faculty of Veterinary Medicine, University of Life Sciences in Lublin, Głęboka St. 30, 20-612 Lublin, Poland; andrzej.zmuda@up.lublin.pl; 6Department and Clinic of Internal Medicine, Faculty of Veterinary Medicine, University of Life Sciences in Lublin, Głęboka St. 30, 20-612 Lublin, Poland; iwona.taszkun@up.lublin.pl; 7Department of Animal Anatomy and Histology, Faculty of Veterinary Medicine, University of Life Sciences in Lublin, Akademicka St. 12, 20-950 Lublin, Poland; karol.rycerz@up.lublin.pl; 8Chair and Department of Paedodontics, Medical University of Lublin, Karmelicka St. 7, 20-081 Lublin, Poland; mielnikmb@gmail.com (M.M.-B.); damian.kuc@umlub.pl (D.K.); 9Department of Animal Physiology, Faculty of Veterinary Medicine, University of Life Sciences in Lublin, Akademicka St. 12, 20-950 Lublin, Poland; ewaRST@interia.pl

**Keywords:** intestine, β-hydroxy-β-methylbutyrate, alpha-ketoglutaric acid, mink, development, prenatal programming

## Abstract

**Simple Summary:**

The American mink has a unique and complex biology, and farmed mink can produce multiple litters. Reproductive success depends on optimal housing, nutrition, body condition and genetic selection. Nutrition during pregnancy affects fetal and offspring development. Therefore, feed supplements must be introduced with caution, especially on high productivity farms, where reproductive efficiency depends on optimal maternal nutrition and maintaining the best possible animal health status. The current study aimed to investigate the structure and maturation of the small intestine in the offspring of primiparous and multiparous mink supplemented with β-hydroxy-β-methylbutyrate and/or alpha-ketoglutaric acid during gestation. Prenatal supplementation induced long-term effects on intestinal development in offspring which were dependent on parity and offspring gender. Intestinal absorption, peristalsis and secretion were affected by prenatal supplementation, as evidenced by the accompanying structural changes. The findings presented here have important nutritional implications, not only for mink breeding but for production overall. The possible effects of the interactions between parity, offspring gender and dietary supplements should be taken into consideration in terms of feeding practice and supplementation plans for the breeding of animals. Further studies are necessary to elucidate possible epigenetic effects of gestational supplementation on the generations of offspring that follow.

**Abstract:**

Prenatal and postnatal supplementation with β-hydroxy-β-methylbutyrate (HMB) and alpha-ketoglutaric acid (AKG) affects the development and maturation of offspring. Both substances have the potential to stimulate cell metabolism via different routes. However, parity affects development and may alter the effects of dietary supplementation. This study aimed to evaluate the effect of gestational supplementation with HMB and/or AKG to primiparous and multiparous minks on the structure and maturation of the offspring’s small intestine. Primiparous and multiparous American minks (*Neovison vison*), of the standard dark brown type, were supplemented daily with HMB (0.02 g/kg b.w.) and/or AKG (0.4 g/kg b.w.) during gestation (*n* = 7 for each treatment). Supplementation stopped when the minks gave birth. Intestine samples were collected from 8-month-old male and female offspring during autopsy and histology and histomorphometry analysis was conducted (LAEC approval no 64/2015). Gestational supplementation had a long-term effect, improving the structure of the offspring’s intestine toward facilitating absorption and passage of intestinal contents. AKG supplementation affected intestinal absorption (enterocytes, villi and absorptive surface), and HMB affected intestinal peristalsis and secretion (crypts and Goblet cells). These effects were strongly dependent on parity and offspring gender. Present findings have important nutritional implications and should be considered in feeding practices and supplementation plans in animal reproduction.

## 1. Introduction

Since feeding is crucial for growth and development, nutritional factors are also important aspects of prenatal growth and prenatal programming. Factors affecting the fetus may have long-lasting outcomes on the further development of selected organs or the organism. These effects can be beneficial or adverse, depending on a variety of factors, including the particular substance in question or environmental circumstances, which in turn can regulate the final developmental outcomes or productivity for instance [1,2,3,4]. High farm productivity is dependent on optimal maternal prenatal nutrient supply and the maintenance of the best possible health status of the animals. These two demands are crucial, specifically in large mink farms. A mother’s diet during pregnancy affects reproductive and rearing periods and offspring development on many levels from the development of key fetal organs, immunity, microbiome composition and function to offspring behavior [5,6,7]. The importance for mink breeding is to minimize economic losses by improving the non-specific and specific immunity of the animals. The goals of general (non-specific) prophylaxis are achieved by providing animals with appropriate feeding and housing conditions. The elimination of stress, nutritional, energy, vitamin and mineral deficiencies has a positive effect on the efficiency of the immune system, and thus reduces the susceptibility of animals to non-infectious and infectious diseases [7,8,9]. However, due to deadlines regarding mating and pelt harvesting and the mean age of dams from outside the basic herd (ca. 6 months) and those in the basic herd (ca. 3 years), the nutritional schemes in large-stock breeding do not allow for individual treatment of the animals. All these factors can result in changes to developmental pathways that provoke further structural alterations in developing organs, like for example in the intestines of offspring, which then further affects their growth [10]. Also, outcomes of particular dietary supplementation during gestation can be affected by many factors, provoking different growth rates in the offspring [11,12,13]. In mink, it is observed that the first litter size is smaller compared to subsequent litters and that the reproductive performance of females increases up to a certain age and then decreases. Thus the parity and the level of milk production in lactating females are also cited among the factors that determine weaning success and kit survival in mink [14]. Information regarding the intrauterine programming of phenotype in farm animals, especially fur animals, is still limited. Any restriction in maternal dietary protein levels decreases fetal protein plasma concentrations, including those of amino acids such as glutamine, proline, alanine, and arginine, which results in further retardation of growth and development [14,15,16]. From this perspective feed additives like β-hydroxy-β-methylbutyrate (HMB) and alpha-ketoglutaric acid (AKG) have potential health benefits. Even though their role in prenatal programming has been assessed, not only in livestock, there are still many unknowns [12,17,18,19,20,21,22,23]. Very limited knowledge about the effects of these amino acid precursors in fur animals exists in literature and there is no information about their effects on the digestive tract of minks. AKG administered enterally is utilized by different bacterial species and intestinal cells and is converted into glutamine, improving protein synthesis. AKG is a precursor of the “glutamate family” of amino acids, whereas, in mammalian cells, glutamine is the key link between carbon metabolism of carbohydrates and proteins and plays an important role in the growth. Glutamine also participates in energy production and, if necessary, in the production of glucose and glycogen [24,25,26]. Moreover, glutamine can serve as an energy source for enterocytes and is used in functional foods for stressful situations such as trauma, cancer, infections, and burns [25,27,28]. On the other hand, HMB, a metabolite of leucine, ameliorates losses in body weight, lean mass, and the reduction of muscle fiber cross-sectional area. It also plays a key role in animal metabolism. Leucine, an essential branched-chain amino acid, acts as a substrate for protein and cholesterol biosynthesis and as a signaling molecule [29]. Animal studies indicate that dietary supplementation with HMB can improve bone and dental development, immune function, overall health and increase the fat content of milk in lactating animals [13,29,30,31]. The farmed mink, compared to its wild counterpart, is characterized by significant changes in gastrointestinal function. A major component of mink feed is by-products from poultry processing and aquatic animals such as fish and shellfish. However, lower nutrient digestibility can adversely affect the reproductive performance of female mink, resulting in higher rates of infertile females and lower birth weights compared to animals fed a diet with optimal digestibility. In addition, mink feed, which is a concentrated source of protein, has a high moisture content and is finely ground, making it an excellent environment for the growth of pathogenic bacteria [14]. Due to the short digestive tract of carnivores and the rapid rate of food passage in the intestines feed intake is largely determined by the quality of gastrointestinal tract structures. Combining the knowledge on animal production and the use of feed supplements seems to be a promising and innovative approach that may have potential application in the treatment of intestinal dysbiosis. The improvement in productivity can be achieved with appropriate additives, which may also adapt the intestines to high protein feed [32,33]. Taking the above into consideration, we hypothesized that dietary supplementation of HMB and AKG to pregnant dams might affect the development and maturation of the gastrointestinal tract in offspring, since both substances have the potential to stimulate cell metabolism, via different routes. However, since primiparity and multiparity may substantially affect the development rate of offspring, we further hypothesized that this factor in mink breeding may alter the effects of the AKG and/or HMG supplementation. Thus, the aim of the current study was to evaluate the effect of gestational supplementation with HMB and/or AKG to primiparous and multiparous minks on the structure and maturation of particular parts of the small intestine in male and female offspring.

## 2. Materials and Methods

### 2.1. Experimental Design

The selection and division of animals into the various experimental groups was done as described previously [19]. The experiment was carried out on clinically healthy, multiparous (after 2–3 pregnancies) and primiparous (at the age of 10 months; had never given birth) American minks (*Neovison vison*), of the standard dark brown type, and their offspring. The animals were housed individually in separate cages, under standard breeding/farming conditions, with natural daylight and free access to fresh water. To obtain an accurate measure of the length of the gestation period, female minks were mated only twice with a fertile male. The mating was performed in March and parturition occurred at the turn of April and May. After the mating, the minks were randomly assigned to one of four groups: a control group (*n* = 14; not supplemented), a group supplemented with AKG (*n* = 14; AKG), a group supplemented with HMB (*n* = 14; HMB), and a group supplemented with HMB + AKG (*n* = 14; HMB + AKG). Each group contained 7 primiparous and 7 multiparous mothers. The feeding, care and breeding of the animals were performed according to routine farm procedures. Mink kits were kept as family groups with their dams until two months of age, after which they were placed singly into separate cages. The animals were fed with standard wet feed based on fish, fish and animal by-products, heat-treated barley, animal fat and rapeseed oil. In particular, the basal diet contained 318 g of dry matter and 168 g crude protein/kg (45.65% of moisture, 35% of protein, 24.15% of fat, 9.43% of carbohydrates, 2.79% of crude fiber, 9.8% of ash, 2.03% of calcium and 1.78% of phosphorus). The metabolizable energy content was 5020 kJ/kg (15.8 MJ/kg DM) [19]. To study the effects of maternal nutrition on the structure and maturation of the small intestines of the offspring, HMB and/or AKG were added to female minks’ feed from day 1 after mating, until the end of the gestation period (ca. 46 days). The AKG-treated groups received alpha-ketoglutaric acid (Protista AB, Sweden) in a daily dose of 0.4 g/kg body weight. The HMB-treated groups received HMB (Lonza, Basel, Switzerland) in a daily dose of 0.02 g/kg body weight. Powdered alpha-ketoglutaric acid was added to the feed following mixing with distilled water and buffering with NaOH to a pH of 7.3, while powdered HMB was added to the feed directly. Controls for the intake of HMB, AKG, and HMB + AKG were performed by mixing the supplements with half of the feed, and giving the animals the feed thus prepared, followed by the second half of the feed after the first half had been consumed. Dosage was estimated according to weekly body weight measurements. In the control group, the first half of the feed and the second half of the feed were administered at the same time as the other groups to avoid introducing another variable into the experiment. The doses of AKG and HMB were based on those used in previous studies [13,26,28,30,31,34]. AKG and HMB supplementation were stopped on the day of parturition and had no impact on reproduction (data not shown) or on the mean rearing rate, which was 4.95. Moreover, the number of live born and stillborn, as well as the number of offspring reared up to the end of the experiment, were not significantly different between the dams in the experimental groups. The division of mink kits to the experimental groups was done according to their mothers’ group origin and history of pregnancies. Each group consisted of 12 mink kits of each sex, which were chosen randomly from the appropriate litters. In summary, there were 4 groups, according to the maternal treatment groups for primiparous and multiparous dams, with 12 males and 12 females. In accordance with Polish legislation and the farm procedures, all the offspring (at the age of 8 months), as well as their mothers were euthanized by carbon monoxide inhalation. The offspring delivered by primiparous and multiparous dams did not differ in final mean body weight, however, male offspring were heavier than females (data not shown). After harvesting, the pelt-skinned carcasses were immediately delivered to the laboratory, where samples of the intestines were collected throughout the autopsy.

### 2.2. Organ Collection and Analyses

Samples of small intestine segments (duodenum, jejunum and ileum) were obtained and processed for histology, as previously described [35,36]. Briefly, 15 mm long segments of the duodenum (2 cm after the stomach), jejunum (from the middle portion of the small intestine) and ileum (4 cm before the ileocolic junction) were collected into a 37 °C saline solution. Samples were placed into a standard histological cassette. Tissues were fixed in 4% buffered formaldehyde (pH 7.0) for 24 h, washed in tap water and dehydrated in graded ethanol solutions (up to 70%), trimmed to allow further cross-section cutting, and processed in Ottix Plus and Ottix Shaper (DiaPath, Martinengo, Italy), with the use of a tissue processor (STP 120, Thermo Scientific, Waltham, MA, USA) to saturate samples in paraffin. Paraffin blocks were then made using an embedding station (MYR EC-350, Casa Álvarez Material Científico S.A., Madrid, Spain). Twenty cross-sections (with 10 mm intervals between each five-slice section), 4 µm thick, were then cut with a microtome (Microm HM 360, Microm, Walldorf, Germany) from each small intestine sample [35,36]. Goldner’s trichrome staining was used to distinguish structures of the intestine wall [36,37]. Bright field microscopic (two-dimensional) images (magnification ×50, ×200 and ×400) were obtained using a confocal microscope (AXIOVERT 200 M, Carl Zeiss, Jena, Germany), with a color digital camera (AxioCam HRc, Carl Zeiss, Jena, Germany) [36]. To analyze the structure of the small intestinal wall, histomorphometry measurements were made on acquired pictures of the small intestine, with the use of graphic analysis software (ImageJ 1.53, National Institutes of Health, Bethesda, MD, USA; available at: http://rsb.info.nih.gov/ij/index.html (accessed on 20 November 2020)). The following parameters were analyzed: mucosa and submucosa thickness; thickness of the inner and outer muscle layer; total thickness of the muscularis; muscle to mucosa, as well as submucosa to mucosa ratio; total crypt number (opened and closed crypts); number of open (showing mitoses and an open internal space, with access to the intestinal lumen) and closed crypts (showing no mitoses and a closed internal space); undamaged, damaged and total number of villi; a ratio of crypt number to villi; number of enterocytes as well as Goblet cells; enterocytes to Goblet cell ratio; enterocyte height (measured as the distance from the brush-border membrane to the basolateral membrane); villi height (from the tip of the villus to the villus–crypt junction) and width (measured in the middle of the villus height); crypt depth (defined as the depth of the invagination between adjacent villi, from the bottom of the crypt to the base of the villus) and width (measured in the middle of the crypt depth) [35,36]. Only vertically oriented villi and crypts were measured. Villi height to crypt depth ratio was calculated. Small intestinal absorptive surface was also determined, according to Kisielinski [38]. A summary of Goldner’s trichrome staining images representative for jejunum can be found in the Appendix A.

### 2.3. Statistical Analysis

Data are expressed as LSMEANS (least-squares mean) ± SEM (standard error of the mean). As our aim was to evaluate the effect of mother type, diet and sex of the offspring on the structure of particular parts of the small intestine, the following statistical model was considered separately for each intestinal segment:x_ijk_ = µ + α_i_ + β_j_ + γ_k_ + (αβ)_ij_ + (αγ)_ik_ + (βγ)_jk_ + (αβγ)_ijk_ + ε_ijkl_
where: x_ijk_—an observation (duodenum, jejunum or ileum parameter value); i—first factor level (diet: HMB, AKG or HMB + AKG); j—second factor level (mother: primiparous or multiparous); k—third factor level (sex: male or female), l—measurement number; μ—constant; α_i_—the main effect of the first factor ith level; β_j_—the main effect of the second factor jth level; γ_k_—the main effect of the third factor kth level; (αβ)_ij_—the effect of interactions between the first and second factor; (αγ)_ik_—the effect of interactions between the first and third factor; (βγ)_jk_—the effect of interactions between the second and third factor; (αβγ)_ijk_—the effect of interactions between the first, second and third factor; ε_ijkl_—random error. Statistical analysis was carried out using an ANOVA for multiple factors. The post hoc Tukey’s honest significant difference test was used as a correction for multiple comparisons. To further determine which groups were different from one another, a contrast analysis was also applied. Normal distribution of data was examined using the W. Shapiro–Wilk test and equality of variance was tested by the Brown–Forsythe test. A two-sided significance level (*p* value) of less than 0.05 was considered statistically significant. A power analysis tool was used to calculate the sample size and the statistical power, as a function of the error. The type and size of the effect for the multiple factors ANOVA was interpreted according to Kotrlik [39]. All statistical analyses were carried out by means of Statistica 13 software (StatSoft, Inc., Tulsa, OK, USA; http://www.statsoft.com (accessed on 21 July 2019)).

## 3. Results

According to the Wilks-λ test, all main effects, as well as all interactions of the variables considered, were statistically significant and had an influence on the structure and maturation of the portions of the small intestine examined (Table 1). However, the most important factor affecting the structure of the small intestine was diet and the Mother×Diet and Diet×Sex interactions (all λ < 0.2), though the Diet×Sex, as well as the Mother×Diet×Sex, interactions were eminent in the jejunum (all λ < 0.1). Over 50% of the total variance was explained by diet and its interactions (big effect size, expressed by the Eta-squared in Table 1). The least crucial factor affecting small intestine structure was sex and its interaction with the type of mother (primiparous or multiparous) (λ > 0.3 for duodenum, λ > 0.4 for jejunum and λ > 0.5 for ileum, Table 1). Detailed changes in the histomorphometric parameters of each of the small intestine sections studied, as well as the comparisons between dietary groups are presented in the tables that follow (duodenum—Table 2; jejunum—Table 3; ileum—Table 4).

### 3.1. Duodenum

#### 3.1.1. Thickness Related Structural Parameters

The majority of parameters referring to the thickness of particular duodenal layers, as well as the calculated thickness proportions, were diminished by maternal HMB, AKG, or both HMB + AKG treatment in male offspring of primiparous dams compared to the control, except for mucosa thickness. Combined HMB + AKG treatment significantly reduced all these thickness parameters, however, only the thickness of the inner muscle layer was diminished by maternal HMB treatment, whereas submucosal thickness and submucosa to mucosa ratio were reduced by AKG treatment. On the contrary, the female offspring of primiparous dams showed almost no response to the maternal treatments, except with regards to the submucosa to mucosa ratio, which was doubled by maternal HMB treatment compared to the control (Table 2). Moreover, there was also a lack of significant structural changes in female offspring of multiparous dams, regardless of treatment, compared to the control, except for the thickness of the mucosa which was significantly increased (by 27%) following maternal HMB treatment (Table 2). For male offspring of multiparous dams, the outer muscle layer and the muscle to mucosa and mucosa to submucosa ratios were elevated compared to the control group following maternal AKG treatment and maternal HMB + AKG treatment. The thickness of the muscularis layer was increased, whereas mucosa thickness was decreased compared to controls, in male offspring of multiparous dams treated with AKG (Table 2). All mentioned parameters related to thickness measurements, except mucosal and partially submucosal thickness, were significant in the main factors studied. Also, significant effects were driven by interactions of main factors, this time not excluding mucosal thickness. In general, most of the parameters relating to wall thickness were higher in offspring from primiparous dams compared to those from multiparous dams, except for mucosa thickness and muscle to mucosa ratio (Appendix A).

#### 3.1.2. Quantity Related Structural Parameters

The effect of the treatments on the number of intestinal crypts and villi was dependent on the type of mother (primiparous versus multiparous). The gestational treatments had no significant impact on the offspring of primiparous dams, irrespective of treatment type, except for a decreased total number of villi observed in females from the HMB + AKG group compared to the controls (Table 3). The gestational treatments had quite the opposite effect on offspring from multiparous dams. Although the total number of crypts in the duodenum of male offspring was not significantly altered by the treatments, prenatal supplementation with HMB, AKG, as well as HMB + AKG increased this parameter in females. There were no significant differences in the number of open crypts in the duodenum of both male and female offspring between treatment groups. All treatments significantly increased the number of closed crypts in female offspring, whereas in male offspring only the HMB treatment increased the number of closed crypts compared to the control and AKG groups (Table 3). The number of damaged villi in the duodenum of female offspring of primiparous minks was increased following HMB treatment compared to the AKG and HMB + AKG groups. The gestational supplementation has a significant impact on the male offspring of multiparous dams, where HMB and HMB + AKG decreased the total number of villi compared to the control group, as well as to the AKG group. However, the number of damaged villi was decreased in all three treatment groups compared to the control. Moreover, significantly more undamaged villi were observed in males from the AKG group compared to those in the HMB group. There were no significant differences in these parameters in the female offspring from multiparous minks (Table 3). Interestingly, even though there were no marked changes in crypt and villi numbers following gestational supplementation in primiparous dams, the crypt to villi ratio was significantly increased in males in the HMB and HMB + AKG treatment groups (Table 3). The opposite effect was observed in the females, where HMB and AKG decreased the crypt to villi ratio compared to the control group. The proportion of crypts to villi was elevated in HMB supplemented groups of offspring males from multiparous minks, and all females, in comparison to respective controls (Table 3). With regards to the number of enterocytes, there were no significant differences in the offspring of primiparous dams, irrespective of gender, compared to the controls. Females from the AKG group however, had 43% more enterocytes per mm of epithelium than that observed in the HMB + AKG group. A substantial increase in enterocyte number was observed in the HMB + AKG males from multiparous dams compared to the control (65%), HMB (72%) and AKG (102%) groups. On the contrary females from the HMB group had 35% fewer enterocytes than offspring in the control group from the same type of mother. Furthermore, males from primiparous dams supplemented with HMB or with AKG had less than half the number of Goblet cells as those from the control group, whereas females from the HMB group had twice as many mucin producing cells than those in the control animals. Interestingly, the results regarding Goblet cell numbers were almost the exact opposite in the offspring from multiparous dams. Prenatal supplementation with AKG doubled the number of Goblet cells in male offspring, while HMB significantly lowered Goblet cell number in females, compared to the respective control groups. The proportion of enterocytes to Goblets cells was significantly increased in male offspring from primiparous dams supplemented with HMB or AKG, compared to the control group. HMB prenatal supplementation decreased the ratio of enterocytes to Goblet cells in female offspring (by 39%) from primiparous dams, compared to controls. In multiparous dams, prenatal supplementation with HMB significantly increased the enterocyte to Goblet cell ratio, whereas AKG supplementation significantly decreased the enterocyte to Goblet cell ratio in male offspring, compared to controls (Table 3). No significant differences in enterocyte to Goblet cell ratio were observed between treatment groups in female offspring from multiparous dams. Interestingly, treatment with AKG or HMB + AKG increased the height of enterocytes in the duodenal epithelium of males from primiparous dams, whereas this effect was not seen in males from multiparous dams. None of the prenatal supplementations had any effect on the height of the enterocytes of female offspring from both primiparous and multiparous dams (Table 3).

#### 3.1.3. Shape and Absorption Surface Related Parameters

Since one of the factors which structurally contributes to mucosa thickness is the villi height, the observed effects of prenatal supplementation with HMB and/or AKG were mostly similar to what was seen with regards to mucosa thickness, with no significant alterations in the villi height of males from primiparous dams and HMB substantially increasing this parameter in females from primiparous dams compared to the control group. AKG supplementation however decreased villi height in males and females from multiparous dams (Table 4). On the contrary, the villi width of female offspring was not affected by prenatal supplementation, regardless of offspring origin. Furthermore, effects exerted by AKG supplementation on male offspring were strongly dependent on the type of mother, since villi width was increased in male offspring from primiparous dams and decreased in male offspring from multiparous dams, compared to their respective controls. HMB + AKG significantly increased crypt width compared to the control group in females from primiparous dams, whereas crypt width was significantly increased by HMB treatment and not significantly decreased by AKG and HMB + AKG treatment in female offspring from multiparous dams, compared to controls. Prenatal supplementation had no significant effects on crypt width or crypt depth, or on the absorptive surface of the duodenum of males from primiparous dams (Table 4). Crypt depth was higher in females from HMB groups, irrespective of the type of mother, compared to respective controls. In the case of the male offspring from multiparous dams, all prenatal treatments decreased the depth of intestinal crypts compared to controls. Villus to crypt ratio was significantly decreased in males from primiparous dams and significantly increased in males from multiparous dams, compared to controls, following HMB treatment. HMB + AKG treatment increased villus to crypt ratio compared to controls in both females and males from multiparous dams. While the absorptive surface of the duodenum was not significantly affected by any of the prenatal supplementations in males from primiparous dams, varied effects were observed in male offspring from multiparous dams. HMB supplementation increased duodenal absorptive surface area and AKG decreased absorptive surface area. HMB treatment in both primiparous and multiparous dams increased absorptive surface area in female offspring, whereas HMB + AKG had a strong stimulating effect on the absorptive surface area in the duodenum of females from multiparous dams (Table 4).

### 3.2. Jejunum

Males from primiparous dams were almost entirely unaffected by prenatal treatments, except for two parameters (Table 5, Table 6 and Table 7, Appendix A).

#### 3.2.1. Thickness Related Structural Parameters

HMB treatment increased the thickness of the mucosa and decreased the thickness of the submucosa, inner muscle layer and muscularis layers in females from primiparous dams. Combined treatment with HMB + HMB + AKG also decreased the thickness of the submucosa, inner muscle layer and muscularis, whereas AKG treatment increased the thickness of the outer muscle layer in females from primiparous dams. In female offspring from multiparous dams, AKG treatment significantly increased the thickness of the mucosa, inner muscle layer, outer muscle layer and the muscularis, as well as the muscle to mucosa ratio, compared to the control group. HMB + AKG treatment increased the submucosa to mucosa ratio in females from multiparous dams. In male offspring from multiparous dams, HMB treatment increased mucosa thickness and decreased the thickness of the inner muscle layer compared to the control group. AKG treatment significantly reduced the thickness of the inner muscle layer and the muscularis, as well as the muscle to mucosa ratio. Combined treatment with HMB + AKG also reduced inner muscle layer thickness compared to controls in males from multiparous dams (Table 5).

#### 3.2.2. Quantity Related Structural Parameters

The number of damaged villi in males from primiparous dams was nearly three times lower in both the HMB and AKG groups, compared to controls, whereas combined treatment had no effect. Additionally, there was a 42% reduction in the number of enterocytes in the HMB group in males from primiparous dams, compared to the control group (Table 6). The number of crypts and villi were not affected by any of the prenatal treatments in males from primiparous dams, except for the number of damaged villi, which as previously mentioned substantially decreased by both HMB and AKG treatments, but not the combined treatment (Table 6). Some of these parameters were altered in females from primiparous dams. Substantially more open crypts were observed in the HMB group and a higher total number of crypts were observed in the AKG group, compared to the control group. Combined prenatal treatment with AKG + HMB significantly elevated the number of closed crypts and decreased the number of damaged villi in female offspring from primiparous dams, compared to controls. All treatments increased the crypt to villi ratio in female offspring of primiparous dams, compared to the control group. All prenatal supplementation to multiparous dams significantly elevated the total number of crypts in male offspring compared to the controls, whereas the total number of crypts remained unchanged in females from multiparous dams. The number of open crypts was elevated in both male and female offspring from multiparous dams following AKG supplementation, whereas combined treatment with AKG + HMB increased the number of closed crypts in males, compared to the control group. HMB supplementation to multiparous dams decreased the number of closed crypts in female offspring, compared to controls. Prenatal supplementation had no effect on the number of undamaged villi in male offspring from multiparous dams, whereas AKG treatment significantly decreased this parameter in females. Nevertheless, AKG elevated the total number of villi and the number of damaged villi, compared to control and HMB groups in male offspring, whereas these parameters were decreased in females by both HMB and AKG treatments, compared to controls. The crypt to villi ratio was higher in males from the HMB and HMB + AKG groups and in females from the AKG group, compared to the control groups. Although gestational supplementation with HMB caused a 42% decrease in enterocyte number in males from primiparous dams, all the treatments almost doubled the number of enterocytes in females from primiparous dams. No significant differences in enterocyte number were observed in males from multiparous dams, whereas at least a 41% increase in enterocyte number was observed in female offspring following AKG supplementation to multiparous dams. Gestational supplementation had no effect on the number of Goblet cells in both male and female offspring or on the enterocyte to Goblet cell ratio in male offspring from primiparous minks. Female offspring from primiparous dams in the HMB + AKG group had less than half the number of Goblet cells compared to those in the AKG group, however, there were no significant alterations compared to the control group. However, the enterocyte to Goblet cell ratio was at least 50% higher in female offspring from primiparous dams, in both the HMB and HMB + AKG groups, compared to the control and AKG groups. In contrast, the number of Goblet cells and the enterocyte to Goblet cell ratio in female offspring from multiparous dams were not affected by prenatal supplementation. In male offspring from multiparous dams, the HMB and HMB + AKG groups had significantly fewer Goblet cells, compared to the control and AKG groups. Thus, the enterocyte to Goblet cell ratios were significantly higher in the HMB and HMB + AKG groups, compared to the control and AKG groups (Table 6).

#### 3.2.3. Shape and Absorption Surface Related Parameters

Even though enterocyte height was not affected by prenatal supplementation or by the type of mother in male offspring, in female offspring from primiparous dams all supplementation groups significantly reduced enterocyte height (by at least 50%) compared to controls. On the other hand, in female offspring from multiparous minks, increased enterocyte height was observed compared to the control group, following HMB and HMB + AKG supplementation (Table 7). Finally, with regards to the villi and crypt dimensions, all prenatal treatments had no significant effects on villi height, villi width, crypt width, crypt depth, villus to crypt ratio, as well as absorption surface in male offspring of primiparous dams. Although there were no significant changes in villi height of female offspring from primiparous dams, the villi width, crypt width and villus/crypt ratio were all substantially decreased in all gestational supplementation groups, compared to controls (Table 7). Crypt depth was significantly elevated by HMB and AKG in females from primiparous dams and in males from multiparous dams. However, in females from multiparous dams, all prenatal treatments significantly reduced crypt depth compared to the control group. Villi height was significantly increased by HMB treatment in males and by HMB treatment and AKG treatment in females from multiparous dams. Villi width was not affected by the prenatal treatments in both male and female offspring from multiparous dams. Crypt width was decreased by AKG treatment in males and increased by HMB treatment and by AKG treatment in females from multiparous dams. Villus to crypt ratio was increased by AKG treatment, while jejunum absorption area was decreased by AKG treatment in male offspring from multiparous dams. In female offspring from multiparous dams, the villus to crypt ratio was significantly increased following prenatal HMB treatment and AKG treatment, whereas the jejunum absorption area was not affected by any of the prenatal treatments (Table 7).

### 3.3. Ileum

Data from histomorphometric examination of the ileum are presented in Table 8, Table 9 and Table 10 and Appendix A.

#### 3.3.1. Thickness Related Structural Parameters

The mucosa thickness of male offspring from primiparous dams was significantly elevated (by 30%) following gestational supplementation with AKG, whereas no significant alterations were noticed in females (Table 8). HMB supplementation to multiparous dams resulted in an increase in mucosa thickness (32%) in male offspring, whereas all of the female treatment groups from multiparous dams had substantially thicker mucosa compared to the control group. Submucosa thickness was not affected by gestational treatments in male offspring from primiparous dams, whereas female offspring from multiparous dams treated with HMB had a thinner submucosa (by 27%) compared to the control group. Furthermore, HMB supplementation reduced the submucosa thickness in male (by 90%) and female offspring (by 69%) from multiparous dams, compared to controls. HMB + AKG treatment and AKG treatment significantly reduced submucosa thickness in male and female offspring from multiparous dams, respectively, compared to their respective control groups The thickness of the inner and outer muscle layers, as well as the total thickness of the muscularis layer were all increased in male offspring by prenatal supplementation with HMB to primiparous (by 101%, 46% and 61%, respectively) and multiparous (by 77%, 61% and 65%, respectively) dams compared to the control groups (Table 8). The thickness of the inner muscle layer and the total thickness of the muscularis layer were not affected by prenatal treatments in female offspring from primiparous dams. However, the thickness of the outer muscle layer in female offspring from primiparous dams supplemented with HMB and with HMB + AKG was significantly lower than that observed in the AKG and control groups. The inner muscle layer of female offspring from multiparous dams was 86% thinner in the AKG group compared to controls and twice as thin as that observed in the control group in the HMB + AKG group. The outer muscle layer and muscularis layer were both thinner in the AKG group of female offspring from multiparous dams compared to the control and HMB groups (Table 8). The muscle to mucosa ratio in the ileum of males from primiparous minks was elevated by prenatal HMB treatment, compared to control and AKG groups. The submucosa to mucosa ratio was significantly reduced in female offspring from primiparous dams treated with HMB and HMB + AKG, compared to the control. Female offspring from multiparous dams in the AKG and HMB + AKG groups had a substantially lower muscle to mucosa ratio compared to the control group, whereas the submucosa to mucosa ratio was decreased by all prenatal treatments, compared to controls. As for the male offspring from multiparous dams, the muscle to mucosa ratio was unaffected by prenatal treatments, whereas the submucosa to mucosa ratio was decreased by prenatal HMB and HMB + AKG treatments, compared to the control group (Table 8).

#### 3.3.2. Quantity Related Structural Parameters

With regards to the number of crypts in the ileum of offspring, the effects of gestational supplementation were strongly dependent on the type of mother (primiparous versus multiparous) (Table 9 and Appendix A). Prenatal treatments had very little effect on crypt and villi numbers in female offspring from primiparous dams, with no changes in the total number of crypts, number of open crypts, number of closed crypts, or the number of undamaged villi observed, compared to the control group (Table 9). The total number of crypts, as well as the number of open crypts, were decreased by HMB treatment and by AKG treatment in male offspring from primiparous dams, whereas the number of closed crypts was unaffected by prenatal treatments. The number of undamaged ileal villi was decreased in male offspring from primiparous dams, whereas the number of damaged villi was decreased by all treatments, compared to the control group. Only prenatal HMB treatment reduced the number of damaged villi in female offspring from primiparous dams. The crypt and villi numbers of male offspring from multiparous dams were not significantly affected by the prenatal treatments, with the total number of crypts, number of open crypts, number of closed crypts, as well as the number of undamaged villi being not significantly different from the control group. HMB treatment and combined treatment with HMB + AKG significantly decreased the total number of crypts and the number of open crypts in female offspring from multiparous dams, whereas all treatments significantly reduced the number of undamaged villi, compared to controls. In male offspring from multiparous dams, all treatments decreased the number of damaged villi, compared to the control group (Table 9). The number of closed crypts was increased by HMB + AKG treatment and the number of damaged villi was decreased by HMB treatment in females from multiparous offspring. HMB supplementation to primiparous dams resulted in a reduced total number of villi in both male and female offspring, compared to the respective control groups. The total number of villi was also reduced in male offspring from primiparous dams treated with AKG. The crypt to villi ratio was unchanged in male offspring from primiparous dams following all prenatal treatments, whereas it was substantially elevated in females from the HMB group, compared to the control group. HMB and HMB + AKG supplementation to multiparous dams significantly decreased the total number of villi, consequently increasing the crypt to villi ratio in male offspring. The total number of villi was reduced in all treatment groups of female offspring from multiparous dams, whereas an increased crypt to villi ratio was only observed in the HMB group, compared to the control group (Table 9). The prenatal treatments exerted a moderate effect on the ileum epithelium, which was less pronounced on the enterocytes than on the Goblet cells. Female offspring from primiparous dams treated with HMB had 59% fewer enterocytes than control animals, whereas those from multiparous dams had almost three times more. Moreover, prenatal supplementation with AKG and HMB + AKG almost doubled the number of enterocytes in female offspring from multiparous dams, compared to control. Similarly in males from multiparous dams, HMB and HMB + AKG increased enterocyte number by 52% and 93%, respectively. The number of Goblet cells was significantly reduced by HMB treatment and AKG treatment in male offspring from primiparous dams and by HMB treatment in female offspring from primiparous dams, compared to the respective control groups. Male offspring from multiparous dams receiving AKG had less than half the number of Goblet cells compared to those in the control group, whereas females offspring from the HMB + AKG group had about 62% more Goblet cells compared to the control group. Consequently, the enterocyte to Goblet cell ratio was twice as high in male offspring from primiparous minks prenatally treated with HMB and AKG, then in control animals, whereas the ratio was decreased (by over one third) in the AKG group of female offspring from primiparous dams. The enterocyte to Goblet cell ratio was increased in male offspring from multiparous dams in the HMB + AKG group, compared to control. All prenatal treatments increased enterocyte to Goblet cell ratio in females from multiparous dams (Table 9).

#### 3.3.3. Shape and Absorption Surface Related Parameters

The height of enterocytes in the ileum of female offspring from primiparous minks was significantly elevated by all prenatal treatments, compared to the control group (Table 10). Enterocyte height in male offspring from both primiparous and multiparous dams, as well as in female offspring from multiparous dams was unaffected by the prenatal treatments. Villi height was increased in male offspring from primiparous dams supplemented prenatally with AKG, compared to the control and HMB + AKG groups. In male offspring from multiparous dams, the HMB + AKG group had the shortest villi, which were significantly shorter than those observed in the AKG group, however, neither were significantly different from the control group. Female offspring from the AKG and HMB + AKG groups had substantially longer villi than the control group (Table 10). With regards to villi width, males from primiparous dams supplemented with HMB and with AKG and males from multiparous dams supplemented with AKG had significantly narrower villi compared to controls. All prenatal supplementation significantly increased the width of villi in multiparous female progeny, compared to the control group (Table 10). Ileum crypts were wider in male offspring from primiparous dams following prenatal supplementation with HMB, AKG and HMB + AKG compared to the control group, with the strongest influence exerted by HMB. HMB supplementation to multiparous dams resulted in wider crypts in male offspring, compared to the control group. The width of the ileal crypts in female offspring from both primiparous and multiparous dams was not affected by prenatal treatments. On the contrary, crypt depth decreased in male offspring from primiparous dams after prenatal HMB + AKG treatment and in male offspring from multiparous dams after prenatal HMB and HMB + AKG treatment, compared to the respective control groups. Crypt depth of female offspring from primiparous dams was increased in the HMB group compared to the control group, whereas HMB treatment decreased crypt depth and AKG treatment increased crypt depth in female offspring from multiparous dams. The villus/crypt ratio was 40% higher in the AKG and HMB + AKG groups of male offspring from primiparous dams, whereas female offspring were not unaffected. On the other hand, villus/crypt ratio was increased in both males and females from multiparous dams treated with HMB and in males in the HMB + AKG group, compared to both the control and AKG groups (Table 10). Finally, the ileal absorptive surface area was significantly increased in female offspring from primiparous supplemented with HMB + AKG and in female offspring from multiparous dams supplemented with AKG and with HMB + AKG, compared to the respective control groups. A significant decrease in absorptive surface area was observed in males from multiparous dams supplemented with HMB and with HMB + AKG, compared to controls (Table 10).

## 4. Discussion

The postnatal development of offspring is influenced by genetic, functional, structural, metabolic and environmental factors that the mother is exposed to during the prenatal period. Complex interactions between these factors also have a substantial effect on adaptive changes in offspring growth and maturation [6,40,41]. Alterations in maternal nutrition or additional feed supplementation throughout the fetal growth period may induce long-term metabolic and systemic effects in postnatal life, which are organ and system dependent [3,42]. The current study focused on the structural adaptations of the gastrointestinal tract in offspring from primiparous and multiparous minks supplemented with HMB and/or AKG during gestation. Although previous studies carried out over the last two decades, based on the above-mentioned substances, have contributed to a broad knowledge base on the effects of HMB and AKG on the skeletal, gastrointestinal, nervous, integumentary, muscular and reproductive systems, only a few have studied the interactions between HMB and AKG and their long-term effects [3,20,21,23,28,30,31,43,44]. However, in these studies, less attention was placed on the gastrointestinal tract and in particular on the prenatal effects of nutritional manipulations on its postnatal development and structure. Additionally, since parity affects reproduction, offspring behavior and maintenance, particularly in fur animals, it has been hypothesized that parity may affect the outcomes of gestational supplementation on offspring development. The results of the current study partly confirmed this hypothesis, except for the fact that neither parity, nor prenatal supplementation significantly influenced reproduction. Other recent studies have shown an association between the occurrence of pre-weaning diarrhea and parity profile, farm size and feeding intensity during the gestational period, on Danish commercial mink farms [45,46]. Since the substances studied in the current study are increasingly used as food supplements for humans and as animal feed additives, knowledge on the possible interaction between parity, offspring gender and type of supplementation could be crucial for human nutrition, as well as for animal breeding and welfare [22,25,34]. Earlier studies have shown a significant stimulatory effect of HMB and AKG on the bodyweight of newborn pigs born from sows supplemented with HMB and/or AKG during the last two or three weeks of pregnancy [13,47]. On the contrary, in the present study, HMB or AKG was given separately or together to pregnant dams (irrespective of parity) did not affect the final body weight of the offspring. Although these data cannot be directly compared, it is clear that different experimental and animal models are still needed to further study the effects of HMB and AKG on offspring postnatal development. To our knowledge, there are no studies on pregnant dams administered simultaneously with HMB and AKG, with regards to the postnatal development of the gastrointestinal tract. Hence it is difficult to directly compare and discuss the present results in the context of the results obtained from previous studies. However, it is apparent that all of the above-mentioned studies indicate that the mammalian gastrointestinal tract of pregnant dams acts as an important system that influences offspring development. Maternal interventions may delay intestine development and thus suppress the digestive and absorptive function of the offspring’s intestine. On the other hand, exposure of the fetus to prenatal maternal microbial components can exert a wide range of both immune and non immune effects on the fetus [41,48]. The immunologically immature intestinal mucosa of the fetus can be prepared by gestational colonization components of the maternal microbiota and direct secretory antibody-mediated protection, which facilitate further colonization of the fetus intestines by endogenous microorganisms [8]. In this manner, AKG given enterally to pregnant dams, which is metabolized by enterocytes and intestinal bacteria, is used as an energy source, which stimulates the digestive tract and maintains pluripotency of embryonic cells [24,25,49]. Although the mechanisms through which AKG affects the cells of the developing fetus are still unclear, it is thought to be either through the direct metabolism of AKG or the indirect transmission of bacterial metabolites, or a combination of both [8,24,49]. HMB is metabolized to HMG-CoA in muscle or mammary tissue, which in turn is used for cholesterol synthesis and influences cell growth and function through the regulation of cell membranes, which also stimulates epithelial cells and promotes offspring performance [3,13,50,51]. Previous studies have shown that HMB supplementation to weaned piglets improved intestinal integrity, function, microbiota communities, and short-chain fatty acid concentrations [52]. The measurements described in the current study were performed in adult offspring, thus allowing us to observe the long-term effects of prenatal supplementation with both HMB and AKG. Surprisingly, the observed effects were stronger than expected and varied depending on parity, offspring sex, supplementation and the interactions between these factors. Since the offspring had no direct alimentary contact with either HMB or AKG, we believe that both of these substances, supplemented to the primiparous and multiparous dams during gestation, exerted strong metabolic or even epigenetic effects at the cellular level, as has recently been confirmed in the case of AKG supplementation [49]. Other previous research has also highlighted the importance of maternal nutrition during the gestational period with regards to offspring development. Differences in postnatal small intestine size and growth/maturation may reflect some variation in the efficiency of maternal nutrient utilization. However, even when apparent tissue growth is unchanged, offspring small intestinal gene expression may be affected by gestational nutrition [53]. In the current study the main focus was on the structural changes in the gut. It is well-known that the number of pregnancies can alter the development and structure of offspring intestines. Certain layers of the intestinal wall were thinner in offspring from multiparous dams, especially in the duodenum and ileum. These changes were also offspring gender dependent. Although, very strong effects of mother type, offspring gender and diet, as well as their interactions, were present in the jejunum parameters, these effects were less pronounced than those observed in the duodenum. The ratio of muscularis thickness to mucosa thickness increased along the intestine portions, in accordance with their functions, with the digestive and absorptive processes declining and peristalsis increasing. This change in proportion was greater in most cases in offspring from multiparous dams, thus the developmental stage of their intestinal structure was more advanced at the end of the study. The prenatal supplementation moderately improved this adaptation, however the effect was offspring gender dependent. Interestingly, even though certain parameters relating to the intestine wall were reduced in offspring from multiparous dams following prenatal treatment, offspring final body weight was unaffected. Thus, the functional efficiency of their intestines must have improved, probably due to the more efficient absorption of nutrients, mainly after supplementation with HMB and AKG. Surprisingly, both treatments administered during gestation to primiparous dams also decreased the structural parameters of the small intestine in most cases. The same applies to the number and shape of the villi and crypts, which in our opinion is characteristic of a more mature intestinal structure and thus a greater functional efficiency in the progeny of multiparous dams. The results obtained from the current study are in accordance with earlier studies and confirm the action of HMB and AKG as metabolism stimulants and energy donors [22,28,30,44]. The effects of HMB gestational supplementation were more abundant and stronger in males than in females, which is also confirmed by other studies which examined the effects of this leucine derivative on organs and body systems [23,43]. Although the HMB effects were less prominent or even inhibitory with regards to certain structural parameters assessed in the intestines of female offspring, the addition of AKG usually ameliorated these effects. Moreover, the inhibitory effect of HMB was more pronounced in offspring from primiparous dams. Both the substances assessed (HMB and AKG) were probably involved in different intestinal functions since AKG affected absorption by altering enterocytes, villi and absorptive surface, especially in the jejunum, whereas HMB altered parameters related to secretion, mainly the crypts and Goblet cells. These results have shown, for the first time, that the addition of HMB or AKG to the diet of pregnant primiparous or multiparous dams exerts long-lasting effects on the intestines of the offspring. To the best of our knowledge, this is also the first study on fur animals, to evaluate the association of maternal diet modulation with postnatal development of the gastrointestinal tract of offspring. The work presented here has some limitations, such as the lack of hormonal analysis of serum and biochemical parameters related to intestinal function. However, the results obtained have important implications for general prevention and show the structural effect of prenatal supplementation and its dependence on parity and sex of offspring. The potential exists here to reduce losses on fur farms through prenatal supplementation with AKG or HMB, which affect the gut and general microbiome and immunology. Another limitation was the use of a single dose of HMB and/or AKG, which was selected on the basis of previous welfare studies in different animal species and therefore cannot be categorically considered the most beneficial in terms of effects on intestinal structure and function in mink. However, the results obtained clearly indicate that AKG and HMB (glutamine precursor and leucine metabolite), can affect intestinal health. The intestinal epithelium is known to play an important role in separating the contents of the intestinal lumen from surrounding tissues. The properties of this barrier are achieved by the formation of a complex multiprotein network between epithelial cells, including tight junctions, adherens junctions and gap junctions. These are essential for maintaining cellular function and homeostasis. Structures for the development of the intestinal absorption surface also play an important role in the physiological functions of the intestine. Further studies are needed in relation to the expression of intestinal barrier proteins and the genes encoding them.

## 5. Conclusions

The present findings have important nutritional implications in terms of mink breeding. The effects of gestational supplementation on offspring development and maturation are dependent on the number of pregnancies, which should be taken into consideration with regards to the feeding practice and supplementation plans for particular animals during breeding. The interaction of the supplements or treatments used should be taken into consideration as they may have certain gender-dependent effects on the postnatal development of the progeny. As for HMB and AKG, further studies should be considered to elucidate the direct or indirect passage of their metabolic influence from dams to the offspring, as well as the possible epigenetic effects on the next generations of progeny.

## Figures and Tables

**Table 1 animals-11-01468-t001:** Wilks multivariate tests of overall significance and effect size.

	Duodenum				Jejunum				Ileum			
	Wilks-λ	F	*p*	Eta-Squared (η2)	Wilks-λ	F	*p*	Eta-Squared (η2)	Wilks-λ	F	*p*	Eta-Squared (η2)
Mother	0.18	29.64	<0.001	0.82	0.33	13.05	<0.001	0.67	0.23	21.07	<0.001	0.77
Diet	0.19	4.64	<0.001	0.42	0.12	6.75	<0.001	0.51	0.10	7.21	<0.001	0.53
Sex	0.38	10.56	<0.001	0.62	0.48	6.95	<0.001	0.52	0.74	2.23	0.002	0.26
Mother×Diet	0.11	7.12	<0.001	0.53	0.15	5.67	<0.001	0.47	0.11	6.94	<0.001	0.52
Mother×Sex	0.31	14.31	<0.001	0.69	0.59	4.45	<0.001	0.41	0.52	5.99	<0.001	0.48
Diet×Sex	0.19	4.68	<0.001	0.42	0.08	8.66	<0.001	0.58	0.17	5.16	<0.001	0.45
Mother×Diet×Sex	0.28	3.40	<0.001	0.35	0.07	9.17	<0.001	0.59	0.24	3.85	<0.001	0.38

β-hydroxy-β-methylbutyrate in the daily dose of 0.02 g/kg of body weight; AKG, alpha-ketoglutaric acid in the daily dose of 0.4 g/kg of body weight; *n*= 12 in each group; significance level at α < 0.05.

**Table 2 animals-11-01468-t002:** Histomorphometric thickness parameters of the duodenum of male (m) and female (f) mink offspring, born from multiparous and primiparous dams, supplemented with HMB and/or AKG during the gestational period.

Mother	Primiparous								Multiparous								SEM
Diet	C		HMB		AKG		HMB + AKG		C		HMB		AKG		HMB + AKG		
Sex	m	f	m	f	m	f	m	f	m	f	m	f	m	f	m	f	
Thickness of the mucosa (µm)	1233	974 _ab_	1109	877_b_	1306	1219 _a_	1127	977 _ab_	1171 ^ac^	964 _b_	1338 ^a^	1228 _a_	591 ^b^	1145 _ab_	984 ^c^	1060 _ab_	21.5
Thickness of the submucosa (µm)	219 ^a^	143	143 ^bc^	169	157 ^b^	152	104 ^c^	130	121	116	110	133	102	122	132	134	3.8
Thickness of the inner muscle layer (µm)	80 ^a^	55	53 ^b^	64	62 ^ab^	59	45 ^b^	58	70 ^ab^	39 _ab_	47 ^b^	44 _ab_	75 ^a^	53 _a_	59 ^ab^	29_b_	1.6
Thickness of the outer muscle layer (µm)	291 ^a^	312	250 ^ab^	266	289 ^a^	239	183 ^b^	241	208 ^b^	144	169 ^b^	162	368 ^a^	140	350 ^a^	151	7.6
Total thickness of muscularis (µm)	373 ^a^	368	304 ^ab^	331	353 ^a^	296	237 ^b^	300	278 ^bc^	183	217 ^c^	207	445 ^a^	183	414 ^ab^	182	8.6
Muscle to mucosa ratio	0.32 ^a^	0.36	0.28 ^ab^	0.37	0.29 ^ab^	0.25	0.20 ^b^	0.34	0.24 ^b^	0.22	0.17 ^b^	0.17	0.96 ^a^	0.16	0.53 ^a^	0.18	0.02
Submucosa to mucosa ratio	0.18 ^a^	0.12 _b_	0.13 ^ab^	0.24_a_	0.13 ^b^	0.14 _ab_	0.09 ^b^	0.15 _ab_	0.11 ^b^	0.14	0.08 ^b^	0.11	0.17 ^a^	0.12	0.15 ^a^	0.13	0.01

Data are presented as means; SEM, standard error of the mean; m, male offspring; f, female offspring; C, control group; HMB, β-hydroxy-β-methylbutyrate at a daily dose of 0.02 g/kg of body weight; AKG, alpha-ketoglutaric acid at a daily dose of 0.4 g/kg of body weight; *n* = 12 in each group; different superscript or subscript letters in rows relating to either the primiparous or multiparous dams refers to a significant difference between males in the various treatment groups or females in the various treatment groups, respectively; (*p* < 0.05).

**Table 3 animals-11-01468-t003:** Histomorphometric quantity parameters of the duodenum of male (m) and female (f) mink offspring, born from multiparous and primiparous dams, supplemented with HMB and/or AKG during the gestational period.

Mother	Primiparous								Multiparous								SEM
Diet	C		HMB		AKG		HMB + AKG		C		HMB		AKG		HMB + AKG		
Sex	m	f	m	f	m	f	m	f	m	f	m	f	m	f	m	f	
Total number of crypts/mm	15.7	19.5	17.9	15.8	17.0	15.9	18.8	15.4	17.2	11.6 _b_	18.3	18.1 _a_	20.3	18.5 _a_	16.3	16.4 _a_	0.3
Number of open crypts/mm	7.9	12.3	8.4	9.6	9.6	9.0	8.7	8.9	12.6 ^ab^	7.6	10.3 ^b^	7.1	18.7 ^a^	10.5	10.2 ^b^	6.7	0.4
Number of closed crypts/mm	7.8	7.3	9.5	6.2	7.4	6.9	8.7	6.5	4.6 ^cb^	4.0 _b_	7.9 ^a^	11.0_a_	1.6 ^c^	8.0 _a_	5.7 ^ab^	9.7 _a_	0.3
Number of undamaged villi/mm	7.9	7.9	7.0	8.1	8.1	8.3	6.9	6.9	9.4 ^ab^	10.3	6.9 ^b^	8.6	11.3 ^a^	7.6	8.9 ^ab^	8.9	0.2
Number of damaged villi/mm	1.0	0.4 _ab_	1.0	1.0 _a_	0.2	0.3 _b_	0.7	0.2 _b_	2.7 ^a^	2.4	0.5 ^b^	2.6	0.8 ^b^	2.6	0.5 ^b^	2.9	0.1
Total number of villi/mm	8.9	8.3 _a_	8.0	9.1 _a_	8.3	8.6 _a_	7.5	6.6 _b_	12.1 ^a^	12.7	7.4 ^b^	11.3	12.1 ^a^	10.5	9.4 ^b^	11.8	0.2
Number of crypts to villi/mm ratio	1.76 ^b^	2.32 _a_	2.27 ^a^	1.76 _b_	2.08 ^ab^	1.87 _b_	2.25 ^a^	2.17 _a_	1.41 ^b^	1.02 _c_	2.48 ^a^	1.57 _ab_	1.68 ^b^	1.78 _a_	1.79 ^b^	1.39 _b_	0.04
Number of enterocytes/mm	117.5	164.3 _ab_	159.5	167.2_ab_	145.9	198.8 _a_	148.4	139.5 _b_	122.1 ^b^	143.7 _a_	117.1 ^b^	93.3 _b_	92.1 ^b^	128.6 _ab_	201.8 ^a^	167.3 _a_	3.9
Number of Goblet cells/mm	47.6 ^a^	14.8 _b_	18.8 ^b^	31.6 _a_	14.7 ^b^	20.8 _b_	29.9 ^ab^	17.3 _b_	21.8 ^b^	17.5 _ab_	12.4 ^b^	12.5 _b_	50.3 ^a^	25.6 _a_	29.5 ^b^	18.4 _ab_	1.2
Enterocytes to Goblet cells ratio	4.1^c^	10.5 _a_	9.5 ^ab^	6.4 _b_	10.5 ^a^	10.6 _a_	6.3 ^bc^	8.4 _ab_	6.7 ^b^	8.7 _ab_	11.2 ^a^	9.9 _a_	2.7 ^c^	5.6 _b_	7.7 ^b^	8.7 _ab_	0.6

Data are presented as means; SEM, standard error of the mean; m, male offspring; f, female offspring; C, control group; HMB, β-hydroxy-β-methylbutyrate at a daily dose of 0.02 g/kg of body weight; AKG, alpha-ketoglutaric acid at a daily dose of 0.4 g/kg of body weight; *n* = 12 in each group; different superscript or subscript letters in rows relating to either the primiparous or multiparous dams refers to a significant difference between males in the various treatment groups or females in the various treatment groups, respectively; (*p* < 0.05).

**Table 4 animals-11-01468-t004:** Histomorphometric shape and absorption surface parameters of the duodenum of male (m) and female (f) mink offspring, born from multiparous and primiparous dams, supplemented with HMB and/or AKG during the gestational period.

Mother	Primiparous								Multiparous								SEM
Diet	C		HMB		AKG		HMB + AKG		C		HMB		AKG		HMB + AKG		
Sex	m	f	m	f	m	f	m	f	m	f	m	f	m	f	m	f	
The height of enterocytes (µm)	22 ^c^	37	24 ^bc^	31	33 ^ab^	41	40 ^a^	38	31 ^ab^	37	36 ^a^	41	45 ^a^	41	20 ^b^	37	1.9
Villi height (µm)	1040	692 _b_	894	1065 _a_	1073	930 _ab_	988	827 _ab_	888 ^ab^	751 _b_	1145 ^a^	990 _a_	520 ^c^	946 _a_	866 ^b^	819 _ab_	32.6
Villi width (µm)	52 ^b^	68	66 ^ab^	63	77 ^a^	71	68 ^ab^	67	81 ^a^	85	68 ^ab^	65	50 ^b^	72	67 ^ab^	73	3.1
Crypts width (µm)	33	30 _b_	37	31 _ab_	35	33 _ab_	36	36 _a_	35	33 _b_	35	37 _a_	32	29 _b_	34	32 _b_	0.7
Crypts depth (µm)	167	118 _b_	193	164 _a_	173	152 _ab_	182	141 _ab_	366 ^a^	395 _b_	162 ^b^	438 _a_	153 ^b^	386 _b_	130 ^b^	280 _b_	19.1
Villus/crypt ratio	6.55 ^a^	5.83	4.76 ^b^	6.71	6.46 ^a^	6.31	5.49 ^ab^	5.80	2.59 ^c^	1.91 _b_	7.31 ^a^	2.19 _b_	3.38 ^c^	2.54 _ab_	5.71 ^b^	2.98 _a_	0.21
Absorption surface (µm^2^)	22.9	19.7 _c_	22.9	30.9 _a_	26.9	26.1 _abc_	25.6	21.8 _bc_	22.0 ^b^	18.9 _c_	29.7 ^a^	25.3 _ab_	11.9 ^c^	27.6 _a_	23.5 ^ab^	22.2 _bc_	1.1

Data are presented as means; SEM, standard error of the mean; m, male offspring; f, female offspring; C, control group; HMB, β-hydroxy-β-methylbutyrate at a daily dose of 0.02 g/kg of body weight; AKG, alpha-ketoglutaric acid at a daily dose of 0.4 g/kg of body weight; *n* = 12 in each group; different superscript or subscript letters in rows relating to either the primiparous or multiparous dams refers to a significant difference between males in the various treatment groups or females in the various treatment groups, respectively; (*p* < 0.05).

**Table 5 animals-11-01468-t005:** Histomorphometric thickness parameters of the jejunum of male and female mink offspring, born from multiparous and primiparous dams, supplemented with HMB and/or AKG during the gestational period.

Mother	Primiparous								Multiparous								SEM
Diet	C		HMB		AKG		HMB + AKG		C		HMB		AKG		HMB + AKG		
Sex	m	f	m	f	m	f	m	f	m	f	m	f	m	f	m	f	
Thickness of the mucosa (µm)	1135	1141 _a_	1226	1002 _b_	1254	1135 _ab_	1131	1142 _a_	1187 ^b^	1010 _bc_	1273 ^a^	1120 _ab_	1208 ^b^	1219 _a_	1002 ^b^	910 _c_	11.9
Thickness of the submucosa (µm)	84	120 _a_	83	77 _c_	101	109 _ab_	94	82 _bc_	109 ^ab^	94	93 ^b^	93	126 ^a^	109	92 ^b^	110	1.9
Thickness of the inner muscle layer (µm)	38	56 _a_	36	35 _b_	40	58 _a_	32	33 _b_	60 ^a^	29 _bc_	41 ^b^	40 _b_	28 ^b^	57 _a_	33 ^b^	22 _c_	1.2
Thickness of the outer muscle layer (µm)	193	171 _b_	150	128 _b_	155	248 _a_	173	139 _b_	153 ^ab^	122 _b_	176 ^a^	140 _b_	115 ^b^	222 _a_	165 ^a^	119 _b_	4.2
Total thickness of muscularis (µm)	232	263 _a_	187	165 _b_	196	309 _a_	200	175 _b_	215 ^a^	154 _b_	217 ^a^	181 _b_	145 ^b^	280 _a_	199 ^a^	145 _b_	5.0
Muscle to mucosa ratio	0.20	0.21 _ab_	0.15	0.17 _b_	0.16	0.28 _a_	0.19	0.16 _b_	0.18 ^a^	0.15 _b_	0.17 ^a^	0.17 _b_	0.12 ^b^	0.24 _a_	0.20 ^a^	0.16 _b_	0.01
Submucosa to mucosa ratio	0.07	0.09 _a_	0.07	0.08 _ab_	0.08	0.10 _a_	0.08	0.07 _b_	0.09 ^ab^	0.09 _b_	0.07 ^b^	0.08 _b_	0.11 ^a^	0.09 _b_	0.10 ^ab^	0.13 _a_	0.002

Data are presented as means; SEM, standard error of the mean; m, male offspring; f, female offspring; C, control group; HMB, β-hydroxy-β-methylbutyrate at a daily dose of 0.02 g/kg of body weight; AKG, alpha-ketoglutaric acid at a daily dose of 0.4 g/kg of body weight; *n* = 12 in each group; different superscript or subscript letters in rows relating to either the primiparous or multiparous dams refers to a significant difference between males in the various treatment groups or females in the various treatment groups, respectively; (*p* < 0.05).

**Table 6 animals-11-01468-t006:** Histomorphometric quantity parameters of the jejunum of male and female mink offspring, born from multiparous and primiparous dams, supplemented with HMB and/or AKG during the gestational period.

Mother	Primiparous								Multiparous								SEM
Diet	C		HMB		AKG		HMB + AKG		C		HMB		AKG		HMB + AKG		
Sex	m	f	m	f	m	f	m	f	m	f	m	f	m	f	m	f	
Total number of crypts/mm	20.3	15.6 _b_	18.4	19.2 _ab_	17.3	20.2 _a_	20.4	19.8 _ab_	14.1 ^b^	17.5	18.7 ^a^	14.7	20.6 ^a^	15.9	19.8 ^a^	20.8	0.3
Number of open crypts/mm	9.8	8.2 _b_	9.1	11.5 _a_	10.5	11.0 _ab_	10.3	9.3 _ab_	7.2 ^b^	4.7 _b_	8.5 ^b^	8.1 _ab_	12.1 ^a^	8.6 _a_	10.0 ^ab^	7.5 _ab_	0.2
Number of closed crypts/mm	10.5	7.4 _b_	9.2	7.6 _b_	7.7	9.2 _ab_	10.1	10.4 _a_	6.9 ^b^	12.8 _a_	9.0 ^ab^	6.0 _b_	8.5 ^ab^	7.3 _ab_	9.9 ^a^	10.9 _ab_	0.3
Number of undamaged villi/mm	8.3	7.6	8.2	8.3	7.2	8.3	8.1	7.4	7.0	8.8 _a_	7.5	7.8 _ab_	8.2	7.0 _b_	7.8	8.1 _ab_	0.1
Number of damaged villi/mm	1.7 ^a^	2.1 _a_	0.6 ^b^	0.6 _b_	0.4^c^	0.4 _b_	1.5 ^ab^	1.2 _ab_	1.7 ^b^	5.1 _a_	1.1 ^bc^	2.0 _b_	4.2 ^a^	1.6 _b_	0.5 ^c^	4.5 _a_	0.1
Total number of villi/mm	10.0	9.8	8.6	8.9	7.8	8.6	9.7	8.6	8.7 ^b^	13.8 _a_	8.7 ^b^	9.8 _b_	12.4 ^a^	8.6 _b_	8.7 ^b^	12.6 _a_	0.2
Number of crypts to villi/mm ratio	2.07	1.64 _b_	2.13	2.17 _a_	2.29	2.35 _a_	2.14	2.32 _a_	1.65 ^b^	1.30 _b_	2.14 ^a^	1.45 _ab_	1.63 ^b^	1.86 _a_	2.37 ^a^	1.66 _ab_	0.04
Number of enterocytes/mm	224.1 ^a^	88.7 _b_	131.1 ^b^	175.9 _a_	181.4 ^ab^	146.3 _a_	210.6 ^a^	170.6 _a_	172.7	190.1 _b_	142.8	170.7 _b_	170.4	268.1 _a_	186.6	155.3 _b_	4.5
Number of Goblet cells/mm	19.1	15.5 _ab_	12.2	17.9 _ab_	16.1	22.3 _a_	20.3	12.8 _b_	21.9 ^a^	17.4	9.9 ^b^	13.8	20.5 ^a^	15.3	13.3 ^b^	15.0	0.6
Enterocytes to Goblet cells ratio	14.36	6.18 _b_	12.51	9.82 _a_	12.60	7.55 _b_	11.37	13.18 _a_	8.32 ^b^	12.35	17.44 ^a^	16.35	8.17 ^b^	17.54	14.18 ^a^	10.30	0.44

Data are presented as means; SEM, standard error of the mean; m, male offspring; f, female offspring; C, control group; HMB, β-hydroxy-β-methylbutyrate at a daily dose of 0.02 g/kg of body weight; AKG, alpha-ketoglutaric acid at a daily dose of 0.4 g/kg of body weight; *n* = 12 in each group; different superscript or subscript letters in rows relating to either the primiparous or multiparous dams refers to a significant difference between males in the various treatment groups or females in the various treatment groups, respectively; (*p* < 0.05).

**Table 7 animals-11-01468-t007:** Histomorphometric shape and absorption surface parameters of the jejunum of male and female mink offspring, born from multiparous and primiparous dams, supplemented with HMB and/or AKG during the gestational period.

Mother	Primiparous								Multiparous								SEM
Diet	C		HMB		AKG		HMB + AKG		C		HMB		AKG		HMB + AKG		
Sex	m	f	m	f	m	f	m	f	m	f	m	f	m	f	m	f	
The height of enterocytes (µm)	49	61 _a_	41	40_b_	40	39 _b_	38	24_c_	36	27 _c_	48	40 _ab_	47	34 _bc_	40	45 _a_	1.1
Villi height (µm)	883	895	1012	840	1027	920	924	845	941 ^bc^	726 _b_	1125 ^a^	967 _a_	999 ^b^	986 _a_	809 ^c^	706 _b_	12.0
Villi width (µm)	82	80 _a_	65	75_b_	82	72 _bc_	64	58 _c_	75	73	82	69	81	81	71	76	1.8
Crypts width (µm)	34	42 _a_	35	36_b_	38	33 _b_	34	32 _b_	37 ^ab^	31 _b_	40 ^a^	48 _a_	26 ^c^	43 _a_	33 ^b^	30 _b_	0.6
Crypts depth (µm)	164.0	142.8 _b_	178.0	184.5 _a_	174.6	181.6 _a_	169.7	168.1 _ab_	163.0 ^c^	294.9 _a_	202.6 ^b^	145.4 _c_	315.3 ^a^	156.1 _c_	152.5 ^c^	229.9 _b_	4.1
Villus/crypt ratio	5.38	6.48 _a_	5.79	4.67 _b_	5.71	5.12 _b_	5.64	5.08 _b_	5.84 ^a^	2.40 _b_	5.65 ^a^	6.57 _a_	3.06 ^b^	6.36 _a_	5.38 ^a^	3.19 _b_	0.11
Absorption surface (µm^2^)	22.7	13.1 _c_	27.0	20.4_b_	24.3	24.9 _ab_	24.6	25.4 _a_	23.3 ^b^	20.5	25.6 ^ab^	19.2	29.2 ^a^	21.0	22.2 ^b^	19.6	0.4

Data are presented as means; SEM, standard error of the mean; m, male offspring; f, female offspring; C, control group; HMB, β-hydroxy-β-methylbutyrate at a daily dose of 0.02 g/kg of body weight; AKG, alpha-ketoglutaric acid at a daily dose of 0.4 g/kg of body weight; *n* = 12 in each group; different superscript or subscript letters in rows relating to either the primiparous or multiparous dams refers to a significant difference between males in the various treatment groups or females in the various treatment groups, respectively; (*p* < 0.05).

**Table 8 animals-11-01468-t008:** Histomorphometric thickness parameters of the ileum of male and female mink offspring, born from multiparous and primiparous dams, supplemented with HMB and/or AKG during the gestational period.

Mother	Primiparous								Multiparous								SEM
Diet	C		HMB		AKG		HMB + AKG		C		HMB		AKG		HMB + AKG		
Sex	m	f	m	f	m	f	m	f	m	f	m	f	m	f	m	f	
Thickness of the mucosa (µm)	727 ^b^	743	846 ^ab^	827	949 ^a^	846	785 ^ab^	790	671 ^b^	562 _b_	889 ^a^	887 _a_	727 ^ab^	770 _a_	748 ^ab^	886 _a_	13.0
Thickness of the submucosa (µm)	102	99 _a_	111	74 _b_	117	99 _a_	105	80 _ab_	243 ^a^	211 _a_	128 ^bc^	125 _b_	221 ^ab^	132 _b_	110 ^c^	180 _ab_	5.6
Thickness of the inner muscle layer (µm)	41 ^b^	48	85 ^a^	39	45 ^b^	40	59 ^b^	39	43 ^b^	95 _a_	76 ^a^	77 _ab_	37 ^b^	51 _b_	62 ^a^	43 _b_	2.0
Thickness of the outer muscle layer (µm)	196 ^b^	221 _b_	287 ^a^	198 _b_	234 ^ab^	279 _a_	263 ^ab^	213 _b_	206 ^b^	306 _a_	332 ^a^	329 _a_	209 ^b^	191 _b_	241 ^b^	241 _ab_	5.9
Total thickness of muscularis (µm)	232 ^b^	269	373 ^a^	239	281 ^bc^	307	323 ^ac^	254	258 ^b^	403 _ab_	425 ^a^	419 _a_	248 ^b^	278 _b_	304 ^b^	285 _b_	7.9
Muscle to mucosa ratio	0.32 ^b^	0.38	0.47 ^a^	0.29	0.30 ^b^	0.37	0.41 ^ab^	0.32	0.38	0.74 _a_	0.50	0.47 _ab_	0.36	0.36 _b_	0.46	0.32 _b_	0.01
Submucosa to mucosa ratio	0.14	0.14 _a_	0.14	0.09 _b_	0.12	0.12 _ab_	0.13	0.10 _b_	0.28 ^a^	0.39 _a_	0.14 ^b^	0.14 _b_	0.32 ^a^	0.17 _b_	0.14 ^b^	0.21 _b_	0.01

Data are presented as means; SEM, standard error of the mean; m, male offspring; f, female offspring; C, control group; HMB, β-hydroxy-β-methylbutyrate at a daily dose of 0.02 g/kg of body weight; AKG, alpha-ketoglutaric acid at a daily dose of 0.4 g/kg of body weight; *n* = 12 in each group; different superscript or subscript letters in rows relating to either the primiparous or multiparous dams refers to a significant difference between males in the various treatment groups or females in the various treatment groups, respectively; (*p* < 0.05).

**Table 9 animals-11-01468-t009:** Histomorphometric quantity parameters of the ileum of male and female mink offspring, born from multiparous and primiparous dams, supplemented with HMB and/or AKG during the gestational period.

Mother	Primiparous								Multiparous								SEM
Diet	C		HMB		AKG		HMB + AKG		C		HMB		AKG		HMB + AKG		
Sex	m	f	m	f	m	f	m	f	m	f	m	f	m	f	m	f	
Total number of crypts/mm	19.9 ^a^	16.9	14.8 ^b^	16.0	15.2 ^b^	17.1	18.5 ^a^	17.0	20.4	23.4 _a_	17.4	17.9 _b_	18.7	18.4 _ab_	19.0	17.8 _b_	0.3
Number of open crypts/mm	13.8 ^a^	9.9	7.7 ^b^	10.1	9.2 ^b^	8.8	11.4 ^ab^	9.8	12.8	18.7 _a_	10.3	11 _b_	11.7	16.8 _a_	10.7	9.0 _b_	0.3
Number of closed crypts/mm	6.1	6.8 _ab_	6.6	5.9_b_	7.4	8.3_a_	7.1	7.2 _ab_	7.6	4.7 _bc_	7.1	7.5 _ab_	7.0	1.6 _c_	8.3	8.8 _a_	0.2
Number of undamaged villi/mm	9.3 ^a^	7.9	7.4 ^b^	6.5	7.7 ^ab^	7.7	9.0 ^ab^	6.7	8.8 ^ab^	15.6 _a_	7.3 ^b^	7.0 _b_	10.1 ^a^	8.7 _b_	8.0 ^ab^	8.1 _b_	0.2
Number of damaged villi/mm	2.1 ^a^	0.7 _a_	0.1 ^b^	0.1 _b_	0.2 ^b^	0.2 _ab_	0.6 ^b^	0.5 _ab_	4.2 ^a^	3.6 _a_	0.5 ^c^	0.4 _b_	2.3 ^b^	2.9 _a_	0.5 ^c^	2.3 _a_	0.1
Total number of villi/mm	11.4 ^a^	8.5 _a_	7.5 ^b^	6.5 _b_	7.9 ^b^	7.3 _ab_	9.4 ^ab^	7.2 _ab_	12.9 ^a^	19.2 _a_	7.8 ^b^	7.4 _b_	12.4 ^a^	11.6 _b_	8.5 ^b^	10.7 _b_	0.3
Number of crypts to villi/mm ratio	1.83	2.07 _b_	2.00	2.52 _a_	2.00	2.24 _ab_	1.98	2.39 _ab_	1.64 ^b^	1.36 _b_	2.23 ^a^	2.50 _a_	1.52 ^b^	1.65 _b_	2.25 ^a^	1.74 _b_	0.04
Number of enterocytes/mm	131.2	213.1 _ab_	139.4	133.7 _c_	110.0	154.3 _bc_	150.7	215.4 _a_	122.4 ^c^	63.6 _c_	185.9 ^b^	180.0 _a_	97.1 ^c^	128.2 _b_	236.2 ^a^	110.0 _b_	4.6
Number of Goblet cells/mm	47.1 ^a^	36.2 _a_	23.4 ^b^	20.8 _b_	23.1 ^b^	34.9 _a_	48.2 ^a^	36.6 _a_	42.0 ^a^	28.8 _b_	34.0 ^a^	36.0 _ab_	20.1 ^b^	40.3 _ab_	37.6 ^a^	46.6 _a_	1.2
Enterocytes to Goblet cells ratio	3.13 ^b^	7.46 _a_	7.15 ^a^	6.19 _ab_	5.75 ^a^	4.58 _b_	3.47 ^b^	5.60 _ab_	3.31 ^b^	2.13 _b_	4.14 ^b^	5.00 _a_	5.01 ^b^	2.76 _a_	7.18 ^a^	2.28 _a_	0.19

Data are presented as means; SEM, standard error of the mean; m, male offspring; f, female offspring; C, control group; HMB, β-hydroxy-β-methylbutyrate at a daily dose of 0.02 g/kg of body weight; AKG, alpha-ketoglutaric acid at a daily dose of 0.4 g/kg of body weight; *n* = 12 in each group; different superscript or subscript letters in rows relating to either the primiparous or multiparous dams refers to a significant difference between males in the various treatment groups or females in the various treatment groups, respectively; (*p* < 0.05).

**Table 10 animals-11-01468-t010:** Histomorphometric shape and absorption surface parameters of the ileum of male and female mink offspring, born from multiparous and primiparous dams, supplemented with HMB and/or AKG during the gestational period.

Mother	Primiparous								Multiparous								SEM
Diet	C		HMB		AKG		HMB + AKG		C		HMB		AKG		HMB + AKG		
Sex	m	f	m	f	m	f	m	f	m	f	m	f	m	f	m	f	
The height of enterocytes (µm)	42	29 _b_	34	35 _a_	36	40 _a_	41	36 _a_	38	38	42	40	39	38	33	37	0.6
Villi height (µm)	561 ^b^	556	607 ^ab^	602	717 ^a^	580	583 ^b^	605	571 ^ab^	459 _c_	542 ^ab^	551 _bc_	616 ^a^	593 _ab_	493 ^b^	685 _a_	8.1
Villi width (µm)	94 ^a^	75 _ab_	70 ^bc^	84 _a_	58 ^c^	89 _a_	87 ^ab^	65 _b_	89 ^a^	35 _c_	80 ^ab^	82 _ab_	61 ^b^	73 _b_	85 ^ab^	95 _a_	1.7
Crypts width (µm)	39 ^b^	38	51 ^a^	38	43 ^a^	39	36 ^a^	36	30 ^b^	38	38 ^a^	38	32 ^ab^	35	37 ^ab^	33	0.5
Crypts depth (µm)	186 ^a^	145 _a_	166 ^a^	175 _b_	174 ^a^	131 _a_	127 ^b^	142 _a_	267 ^a^	256 _b_	134 ^b^	143 _c_	287 ^a^	329 _a_	125 ^b^	278 _b_	5.3
Villus/crypt ratio	3.04 ^b^	3.88	3.69 ^ab^	3.54	4.27 ^a^	4.49	4.54 ^a^	4.41	2.17 ^b^	1.83 _b_	4.09 ^a^	3.85 _a_	2.22 ^b^	1.78 _b_	3.84 ^a^	2.42 _b_	0.09
Absorption surface (µm^2^)	12.6	13.0 _b_	12.4	14.1 _ab_	15.3	13.3 _ab_	14.2	16.1 _a_	14.9 ^a^	11.6 _c_	13.1 ^b^	13.1 _bc_	17.2 ^a^	15.4 _ab_	11.8 ^bc^	16.6 _a_	0.2

Data are presented as means; SEM, standard error of the mean; m, male offspring; f, female offspring; C, control group; HMB, β-hydroxy-β-methylbutyrate at a daily dose of 0.02 g/kg of body weight; AKG, alpha-ketoglutaric acid at a daily dose of 0.4 g/kg of body weight; *n* = 12 in each group; different superscript or subscript letters in rows relating to either the primiparous or multiparous dams refers to a significant difference between males in the various treatment groups or females in the various treatment groups, respectively; (*p* < 0.05).

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
