# Peer review of "The Effects of Prenatal Supplementation with β-Hydroxy-β-Methylbutyrate and/or Alpha-Ketoglutaric Acid on the Development and Maturation of Mink Intestines Are Dependent on the Number of Pregnancies and the Sex of the Offspring"

_animals, 2021, doi:10.3390/ani11051468_

Round 1
Reviewer 1 Report
The Authors ahve performed an interesting study aimed on the analysis of the maternal supplementation during pregnancy with β-hydroxy-β-methylbutyrate (HMB) and 2-oxoglutaric acid (2-Ox) and its effect on the intestinal histology in their offspring. The presented study is well designed and appropriate methods were used to examine the hypothesis. The results are clearly presented and well discussed. The concept of the study is actual and of a great importance from the point of view of nutritional development programming which can exert not only the short-term effects but also influence the animal phenotype and future productivity. Instead of the fur production, which is getting controversial in the EU countries from ethical and alleged sars-CoV2 transmission issues, the results obtained on the American mink model used in the study may be useful for the nutritional programing analysis in other carnivora (companion animals) but also in other animal species including farm animals. Thus, I am recommending this paper for publication in Animals.
In my opinion the further studies concerning analysis of the epigenetic changes in the intestinal mucosa after the analyzed supplementation should be considered in the future studies.
Author Response
Response: We kindly thank the reviewer for such positive comments and appreciation of our work.
After discussion with the co-authors, we decided to change the nomenclature used for the glutamine derivative from 2-oxoglutaric acid (2-Ox) to alpha-ketoglutaric acid (AKG) according to the European Food Safety Authority nomenclature.
Reviewer 2 Report
In this study, the authors investigated the structure and maturation of the small intestine in the offspring of primiparous and multiparous mink supplemented with β-hydroxy-β-methylbutyrate and/or 2-oxoglutaric acid during gestation. Prenatal supplementation induced long-term effects on intestinal development in offspring which were dependent on parity and offspring gender. Intestinal absorption, peristalsis and secretion were affected by prenatal supplementation, as evidenced by the accompanying structural changes. The findings present here have important nutritional implications, not only for mink breeding, but for production overall. The possible effects of the interactions between parity, offspring gender and dietary supplements should be taken into consideration in terms of feeding practice and supplementation plans for the breeding of particular animals. During the review, there are several questions that were raised.
In line 53-56, use “alter” to describe the results in the Abstract is not clear. Use “decrease” “increase” “enhance” et al to clearly describe the changes.
“These effects were strongly dependent on parity and offspring gender” the description is also blurry. Describe specifically how the effects of β-hydroxy-β-methylbutyrate and/or 2-oxoglutaric acid on intestine dependent on parity and offspring gender.
What does “overall intestinal efficiency” mean?
The author described that HMB and/or 2-Ox were added to female minks’ feed in a daily dose of 0.02 g/kg body weight and 0.4 g/kg body weight. Did the minks have a controlled feed intake every day? Please add the data of feed intake.
Representative pictures of Goldner’s trichrome staining should be included in the manuscript.
Author Response
Reviewer 2:
In this study, the authors investigated the structure and maturation of the small intestine in the offspring of primiparous and multiparous mink supplemented with β-hydroxy-β-methylbutyrate and/or alpha-ketoglutaric acid during gestation. Prenatal supplementation induced long-term effects on intestinal development in offspring which were dependent on parity and offspring gender. Intestinal absorption, peristalsis and secretion were affected by prenatal supplementation, as evidenced by the accompanying structural changes. The findings present here have important nutritional implications, not only for mink breeding, but for production overall. The possible effects of the interactions between parity, offspring gender and dietary supplements should be taken into consideration in terms of feeding practice and supplementation plans for the breeding of particular animals. During the review, there are several questions that were raised.
Response: After discussion with the co-authors, we decided to change the nomenclature used for the glutamine derivative from 2-oxoglutaric acid (2-Ox) to alpha-ketoglutaric acid (AKG) according to the European Food Safety Authority nomenclature.
In line 53-56, use “alter” to describe the results in the Abstract is not clear. Use “decrease” “increase” “enhance” et al to clearly describe the changes.
Response: We have revised the sentences to read: “Gestational supplementation had a long-term effect, improving the structure of the offspring's intestine toward facilitating absorption and passage of intestinal contents. AKG supplementation affected intestinal absorption (enterocytes, villi and absorptive surface), and HMB affected intestinal peristalsis and secretion (crypts and Goblet cells).”
“These effects were strongly dependent on parity and offspring gender” the description is also blurry. Describe specifically how the effects of β-hydroxy-β-methylbutyrate and/or alpha-ketoglutaric acid on intestine dependent on parity and offspring gender.
Response: We agree with the reviewer that this description may sound vague, but we are limited by the number of words in the abstract, and this statement addresses the effect size, the differences in structural responses between males and females, as well as the impact of parity. This was the shortest way to highlight the phenomenon and keep the reader interested.
What does “overall intestinal efficiency” mean?
Response: For better clarity we have revised the sentence to read: “Gestational supplementation had a long-term effect, improving the structure of the offspring's intestine toward facilitating absorption and passage of intestinal contents.”
The author described that HMB and/or AKG were added to female minks’ feed in a daily dose of 0.02 g/kg body weight and 0.4 g/kg body weight. Did the minks have a controlled feed intake every day? Please add the data of feed intake.
Response: Controls for the intake of HMB, AKG, and HMB+AKG were performed by mixing the supplements with half of the feed and giving the animals the feed thus prepared, followed by the second half of the feed after the first half had been consumed. In the control group, the first half of the feed and the second half of the feed were administered at the same time as the other groups to avoid introducing another variable into the experiment. More detailed description has been added to the material and methods section. (Lines 167-172)
Representative pictures of Goldner’s trichrome staining should be included in the manuscript.
Response: Although we agree with the reviewer, it was very difficult for us to select microscopic images that would relate to the changes revealed by the histomorphometric analysis and present them in a form that would be clearly distinguishable to the reader. Because the animals in the experiment were healthy, no malformations or pathological changes were found in the intestines, as confirmed by two experienced histologists. 192 microscopic images from each group were analyzed (from 16 groups, for a total of more than 3000 images). Histomorphometry is an established method for distinguishing structural differences that are not observable "at a glance". Providing clearly visible differences that would be representative of the results obtained would force us to show maxima and minima, which would also be a manipulation of the data. Therefore, after discussion with all co-authors, we concluded that figures of intestinal cross-sections would be added as supplementary material. We hope that our decision will be understood by the esteemed reviewer. The sentence was added to the materials and method section (lines 222-224) “A summary of Goldners trichrome staining images representative for jejunum can be found in the supplementary materials as Figure 1S.”
Reviewer 3 Report
Major comments
Introduction part: it is missing an elaboration on the other factors than the dietary treatment that were used in the statistical model about their potential effects on growth and development of the intestine morphology parameters evaluated in the present study. Why for example a classification was made between primiparous and multiparous mink reproducing females? The importance of these factors should be briefly explained and justified.
Lines 72-73: “metabolic diseases during the reproductive and rearing periods, as well as diseases in the perinatal period”. The authors should be more specific to which metabolic or not diseases they are referring to in minks and provide a rationale how the tested ingredients can help towards this direction. This is only addressed as a hypothesis in lines 104-107, however it should have been better clarified why the supplementation of these ingredients are of specific importance for the minks. Moreover, it is a fur animal, and a link between these ingredients and fur production and quality is missing.
Lines 236-343: This section requires revision in the way findings are presented. There is a plethora of findings, which the way they are written and also presented in the Table 2, makes it rather difficult for the reader to understand, distinguish and digest the findings. It is suggested to the authors to write differently this part by splitting in paragraphs or in numbered subsections, and secondly to present some significant parameters of Table 2 in graphs (e.g. bar graphs). Or even present the data in more separate tables. Some figures depicting e.g. mucosal thickness differences between treatments could also be useful.
Lines 350-435, 442-562: the same remark as last comment. Authors are advised to make the necessary changes.
Minor comments
Lines 27-32: these lines should be more focused to the animal specie investigated, and/or to polytocous and multiparous species in more general way. These species are of specific needs during gestation and lactation.
Line 48: provide the number of minks-replicates per treatment
Line 57-58: last sentence is rather vague. Should be more focused on the specie investigated.
Line 82: this protein restriction can be common in the field? Or it has been studied previously in minks?
Lines 160-161: the authors need to mention clearly when the sampling occurred and not only as a citation.
Line 344: male and female add brackets (m) and (f)
Lines 630-633: should be revised, as only one level of each ingredient was used in the experiment, and how strong the effect was can be a bit subjective. Authors should acknowledge the limitation of their research.
Author Response
Response: After discussion with the co-authors, we decided to change the nomenclature used for the glutamine derivative from 2-oxoglutaric acid (2-Ox) to alpha-ketoglutaric acid (AKG) according to the European Food Safety Authority nomenclature.
Major comments
Introduction part: it is missing an elaboration on the other factors than the dietary treatment that were used in the statistical model about their potential effects on growth and development of the intestine morphology parameters evaluated in the present study. Why for example a classification was made between primiparous and multiparous mink reproducing females? The importance of these factors should be briefly explained and justified.
Response: The information on other factors is added to read: “In mink, it is observed that the first litter size is smaller compared to subsequent litters, and that the reproductive performance of females increases up to a certain age and then decreases. Thus, the parity and the level of milk production in lactating females is also cited among the factors that determine weaning success and kit survival in mink [14]”. As well as rationale to the potential on growth and development and structural studies: “The farmed mink, compared to its wild counterpart, is characterized by significant changes in gastrointestinal function. A major component of mink feed is by-products from poultry processing and aquatic animals such as fish and shellfish. However, low-er nutrient digestibility can adversely affect the reproductive performance of female mink, resulting in higher rates of infertile females and lower birth weights compared to animals fed a diet with optimal digestibility. In addition, mink feed, which is a concentrated source of protein, has a high moisture content and is finely ground, making it an excellent environment for the growth of pathogenic bacteria [14]. Due to the short digestive tract of carnivores and the rapid rate of food passage in the intestines feed in-take is largely determined by the quality of gastrointestinal tract structures. Combining the knowledge on animal production and the use of feed supplements seems to be a promising and innovative approach which may have potential application in the treatment of intestinal dysbiosis. The improvement in productivity can be achieved with appropriate additives, which may also adapt the intestines to high protein feed [32,33]”.
Lines 72-73: “metabolic diseases during the reproductive and rearing periods, as well as diseases in the perinatal period”. The authors should be more specific to which metabolic or not diseases they are referring to in minks and provide a rationale how the tested ingredients can help towards this direction. This is only addressed as a hypothesis in lines 104-107, however it should have been better clarified why the supplementation of these ingredients are of specific importance for the minks. Moreover, it is a fur animal, and a link between these ingredients and fur production and quality is missing.
Response: We have revised the sentences to read: “A mother's diet during pregnancy affects reproductive and rearing periods and off-spring development on many levels from development of key fetal organs, immunity, microbiome composition and function to offspring behavior”.” The important for mink breeding is to minimize economic losses by improving the non-specific and specific immunity of the animals. The goals of general (non-specific) prophylaxis are achieved by providing animals with appropriate feeding and housing conditions. The elimination of stress, nutritional, energy, vitamin and mineral defi-ciencies has a positive effect on the efficiency of the immune system, and thus reduces the susceptibility of animals to non-infectious and infectious diseases”. Dear Reviewer, based on our previous studies in various animal models, we have gained data and knowledge on the effects of HMB and AKG on reproduction, digestive and skeletal systems, but to the best of our knowledge we have not been able to find studies where these substances have been directly studied or linked to fur animal production. Therefore, our goal was to try to incorporate this knowledge into fur animal production. Out thinking direction was that there are four phases in mink nutrition, depending on the physiological state associated mainly with reproduction. The first phase is the preparation of animals for reproduction, aimed at reducing body weight by 10%. The second phase is the gestation period, during which feeding is similar to the preparation phase. And here HMB and/or AKG were administered. The most difficult period of feeding is lactation and rearing of young, because the nutritional needs increase significantly, and the change of ration from the previous one to the pregnancy one cannot be made abruptly, because gastrointestinal disorders may occur. The next stage is feeding during the formation of the winter coat. The most important thing in mink breeding is to minimize economic losses by improving the non-specific and specific immunity of the animals. The goals of general (non-specific) prophylaxis are achieved by providing animals with appropriate feeding and housing conditions. The elimination of stress, nutritional, energy, vitamin and mineral deficiencies has a positive effect on the efficiency of the immune system, and thus reduces the susceptibility of animals to non-infectious and infectious diseases. The primary goal in preventing mink farming is to eliminate food poisoning that occurs due to the use of available slaughter waste from animal production. From the point of view of general prophylaxis and non-specific immunity, it is also important to add to feed preparations that protect parenchymal organs, preparations that degrade and inactivate mycotoxins, which enables better results in breeding and reproduction of carnivorous fur-bearing animals. Nutritional supplements supplement the basic feed composition and enhance the immune status of the body. Nutritional supplements are being sought to increase not only dry matter intake of feed, but more importantly to reduce lipolysis of spare fat and reduce morbidity. Many other previously conducted studies suggest that AKG and HMB may be such supplements. In addition, although it is known that the aforementioned formulations do not comprehensively address the problem associated with mycotoxins, but by improving intestinal epithelial structures may minimize liver and gastrointestinal damage caused by mycotoxins present in the feed, obviously further research is required. And in this point, we were trying to study the effects of AKG and HMB on the mink model with consideration of parity and offspring gender to further study these substances in next steps of more prophylactic use. The results of our present study are important for designing the next steps in our research. The widespread occurrence of many immunosuppressive factors in the environment of carnivorous fur-bearing animals makes specific immunoprophylaxis against many infectious diseases difficult, prompting the use of other effective methods to prevent their adverse effects on the immune system of the organism. The possibility of restoring the disturbed homeostasis of the organism creates the basis for the use of a variety of approaches in the prevention and treatment of infectious diseases in carnivorous fur-bearing animals. Such a role is played by the control of the immune system, especially in terms of practical use (administration to mothers, and the effects obtained in offspring), as indicated by many years of research using AKG and HMB. Non-specific immunomodulation using natural or synthetic immunomodulators, which include AKG and HMB, is increasingly used in the prevention and treatment of many diseases.
Lines 236-343: This section requires revision in the way findings are presented. There is a plethora of findings, which the way they are written and also presented in the Table 2, makes it rather difficult for the reader to understand, distinguish and digest the findings. It is suggested to the authors to write differently this part by splitting in paragraphs or in numbered subsections, and secondly to present some significant parameters of Table 2 in graphs (e.g. bar graphs). Or even present the data in more separate tables. Some figures depicting e.g. mucosal thickness differences between treatments could also be useful. Lines 350-435, 442-562: the same remark as last comment. Authors are advised to make the necessary changes.
Response: The changes to the tables presentation were made and the division of the text into the paragraphs
Minor comments
Lines 27-32: these lines should be more focused to the animal specie investigated, and/or to polytocous and multiparous species in more general way. These species are of specific needs during gestation and lactation.
Response: We have revised the sentences to read: “The American mink has a unique and complex biology, and farmed mink can produce multiple litters. Reproductive success depends on optimal housing, nutrition, body condition and genetic selection. Nutrition during pregnancy affects fetal and offspring development”.
Line 48: provide the number of minks-replicates per treatment
Response: We have revised the sentences to read: “Primiparous and multiparous American minks (Neovison vison), of the standard dark brown type, were supplemented daily with HMB (0.02 g/kg b.w.) and/or AKG (0.4 g/kg b.w.) during gestation (n=7 for each treatment)”.
Line 57-58: last sentence is rather vague. Should be more focused on the specie investigated.
Response: We have revised the sentences to read: “Present findings have important nutritional implications and should be considered in feeding practices and supplementation plans in animal reproduction”.
Line 82: this protein restriction can be common in the field? Or it has been studied previously in minks?
Response: Although this protein restriction is studied in the field, we have also added a reference according to mink studies. (Ludwiczak, A.; Stanisz, M. The reproductive success of farmed American mink (Neovison vison) - A review. Annals of Animal Science 2019, 19, 273–289)
Lines 160-161: the authors need to mention clearly when the sampling occurred and not only as a citation.
Response: Dear Reviewer, please forgive us if we do not fully understand this comment. These citations refer to our other studies that used histological techniques. This is to allow the reader to follow possible changes in the overall technique, which may depend on the animal model and organ studied, the equipment, and the particular techniques.
Line 344: male and female add brackets (m) and (f)
Response: Brackets are added
Lines 630-633: should be revised, as only one level of each ingredient was used in the experiment, and how strong the effect was can be a bit subjective. Authors should acknowledge the limitation of their research.
Response: We have revised the sentences to read “Since the offspring had no direct alimentary contact with either HMB or AKG, we believe that both of these substances, supplemented to the primiparous and multiparous dams during gestation, exerted metabolic or even epigenetic effects at the cellular level, as has recently been confirmed in the case of AKG supplementation.” The limitations of our study are now included to the end part of the discussion: ”The work presented here has some limitations, such as the lack of hormonal analysis of serum and biochemical parameters related to intestinal function. However, the results obtained have important implications for general prevention and show the structural effect of prenatal supplementation and its dependence on parity and sex of offspring. The potential exists here to reduce losses on fur farms through prenatal supplementation with AKG or HMB, which affect the gut and general microbiome and immunology. Another limitation was the use of a single dose of HMB and/or AKG, which was selected on the basis of previous welfare studies in different animal species and therefore cannot be categorically considered the most beneficial in terms of effects on intestinal structure and function in mink. However, the results obtained clearly indicate that AKG and HMB (glutamine precursor and leucine metabolite), can affect intestinal health. The intestinal epithelium is known to play an important role in separating the contents of the intestinal lumen from surrounding tissues. The properties of this barrier are achieved by the formation of a complex multiprotein network between epithelial cells, including tight junctions, adherens junctions and gap junctions. These are essential for maintaining cellular function and homeostasis. Structures for the development of the intestinal absorption surface also play an important role in the physiological functions of the intestine. Further studies are needed in relation to the expression of intestinal barrier proteins and the genes encoding them”.
Round 2
Reviewer 3 Report
The authors have complied with the majority of the suggested changes/modifications. The rationale of the study is better described and the limitations of the study are also presented in an objective manner.